# Glucagon-like peptide-1 receptor activation stimulates PKA-mediated phosphorylation of Raptor and this contributes to the weight loss effect of liraglutide

**Thao DV Le[1], Dianxin Liu[2], Gai-Linn K Besing[1], Ritika Raghavan[1], Blair J Ellis[1], Ryan P Ceddia[2], Sheila Collins[1,2]\*, Julio E Ayala[1]\***

[1]Department of Molecular Physiology and Biophysics, Vanderbilt University School of Medicine, Nashville, United States; [2]Department of Medicine, Vanderbilt University Medical Center, Nashville, United States

**\*For correspondence:**
sheila.collins@vumc.org (SC);
julio.e.ayala@vanderbilt.edu
(JEA)

**Competing interest:** The authors declare that no competing interests exist.

**Abstract** The canonical target of the glucagon-like peptide-1 receptor (GLP-1R), Protein Kinase A (PKA), has been shown to stimulate mechanistic Target of Rapamycin Complex 1 (mTORC1) by phosphorylating the mTOR-regulating protein Raptor at Ser[791] following β-adrenergic stimulation. The objective of these studies is to test whether GLP-1R agonists similarly stimulate mTORC1 via PKA phosphorylation of Raptor at Ser[791] and whether this contributes to the weight loss effect of the therapeutic GLP-1R agonist liraglutide. We measured phosphorylation of the mTORC1 signaling target ribosomal protein S6 in Chinese Hamster Ovary cells expressing GLP-1R (CHO-Glp1r) treated with liraglutide in combination with PKA inhibitors. We also assessed liraglutide-mediated phosphorylation of the PKA substrate RRXS\*/T\* motif in CHO-Glp1r cells expressing Myc-tagged wild-type (WT) Raptor or a PKA-resistant (Ser[791]Ala) Raptor mutant. Finally, we measured the body weight response to liraglutide in WT mice and mice with a targeted knock-in of PKA-resistant Ser[791]Ala Raptor. Liraglutide increased phosphorylation of S6 and the PKA motif in WT Raptor in a PKA-dependent manner but failed to stimulate phosphorylation of the PKA motif in Ser[791]Ala Raptor in CHO-Glp1r cells. Lean Ser[791]Ala Raptor knock-in mice were resistant to liraglutide-induced weight loss but not setmelanotide-induced (melanocortin-4 receptor-dependent) weight loss. Diet-induced obese Ser[791]Ala Raptor knock-in mice were not resistant to liraglutide-induced weight loss; however, there was weight-dependent variation such that there was a tendency for obese Ser[791]Ala Raptor knock-in mice of lower relative body weight to be resistant to liraglutide-induced weight loss compared to weight-matched controls. Together, these findings suggest that PKA-mediated phosphorylation of Raptor at Ser[791] contributes to liraglutide-induced weight loss.

## Editor's evaluation

This manuscript provides support for the importance of PKA-dependent mTORC1 activation for the weight-loss effects of Liraglutide, and presumably many GPCR-dependent anorectic agents, while not affecting the food intake-reducing effects of the MC4Ractivator setmelanotide. The weight loss effect of liraglutide however was not impaired after HFD feeding, a model of obesity where GLP1 agonists are being used.

## Introduction

Glucagon-like peptide-1 (GLP-1) receptor (GLP-1R) agonists reduce food intake and body weight and are approved for weight loss (*Drucker, 2022*). Elucidating the mechanisms through which GLP-1R agonists regulate body weight could enhance the therapeutic potential of these compounds or identify novel targets for anti-obesity drug development. We have shown that the GLP-1R agonist Exendin-4 (Ex4) stimulates mechanistic Target of Rapamycin (mTOR) Complex-1 (mTORC1) signaling and that inhibition of mTORC1 signaling in the ventromedial hypothalamus (VMH) with rapamycin blocks the anorectic effect induced by Ex4 in this nucleus (*Brown et al., 2018*; *Burmeister et al., 2017*). The present studies aim to define the mechanism(s) by which GLP-1R agonism stimulates mTORC1 signaling and reduces body weight.

mTOR is a Ser/Thr protein kinase that forms two complexes: mTORC1 and mTOR complex 2 (mTORC2; *Saxton and Sabatini, 2017*). The rapamycin-sensitive mTORC1 has unique accessory proteins including Raptor (regulatory associated protein of mTOR). mTORC1 activity is stimulated by growth factors, nutrients, and hormones by promoting PI3K/Akt-mediated phosphorylation of tuberous sclerosis complex 1/2 (TSC1/2) (*Kim et al., 2002*). Activation of mTORC1 by GLP-1R agonists seemed unlikely since the GLP-1R is a $G_{\alpha}s$-coupled receptor that signals via protein kinase A (PKA; *Thorens, 1992*). However, activation of the $G_{\alpha}s$-coupled β-adrenergic receptor stimulates mTORC1 via Raptor phosphorylation by PKA (*Liu et al., 2016*), suggesting that GLP-1R activation can similarly stimulate mTORC1 via PKA phosphorylation of Raptor.

Here we demonstrate that GLP-1R activation induces mTORC1 signaling and Raptor phosphorylation in a PKA-dependent manner and show that this contributes to weight loss by the therapeutic GLP-1R agonist liraglutide. This reveals a novel weight-lowering GLP-1R-PKA-mTORC1 axis and broadly suggests that other $G_{\alpha}s$-coupled receptors may use this PKA-mTORC1 pathway for certain biological responses.

## Results

### GLP-1R agonists activate mTORC1 in a PKA-dependent manner

Similar to Ex4 (*Brown et al., 2018*; *Burmeister et al., 2017*), liraglutide increases phosphorylation of the mTORC1 signaling target ribosomal protein S6 in CHO-GLP-1R cells, and this is blocked by the mTORC1 inhibitor rapamycin (*Figure 1A*; *Figure 1—figure supplement 1*). The PKA inhibitor H89 also blocks S6 phosphorylation by liraglutide or the adenylyl cyclase/PKA activator forskolin (*Figure 1B*; *Figure 1—figure supplement 1*). However, H89 has some off-target effects including inhibiting the S6 kinase p70S6K (*Davies et al., 2000*), likely explaining why H89 inhibits S6 phosphorylation by insulin (*Figure 1B*). KT5720, a PKA inhibitor with minimal off-target effects on p70S6K (*Davies et al., 2000*), also reduces liraglutide- and forskolin-induced phosphorylation of S6 and the PKA target CREB in a dose-dependent manner (*Figure 1C*; *Figure 1—figure supplement 1*). Neither dose of KT5720 affected S6 phosphorylation by insulin (*Figure 1C*). Two doses of the pan-Akt inhibitor MK-2206 that block insulin-stimulated Akt phosphorylation did not affect stimulation of mTORC1 activity by liraglutide (*Figure 1D*; *Figure 1—figure supplement 1*). These data suggest that liraglutide stimulates mTORC1 signaling in a PKA-dependent and Akt-independent manner.

### GLP-1R activation induces PKA phosphorylation of Raptor at Ser[791]

The catalytic subunit of PKA directly phosphorylates Ser[791] of Raptor (*Liu et al., 2016*). We tested whether liraglutide promotes PKA phosphorylation of Raptor at Ser[791] by expressing Myc-tagged wild-type (WT) Raptor or Myc-tagged mutant Raptor with alanine replacing serine at the 791st position (Ser[791]Ala Raptor) in CHO-GLP-1R cells. Using an antibody against the PKA motif RRXS*/T*, immunoblotting of immunoprecipitated Myc-proteins shows that liraglutide and forskolin increase PKA motif phosphorylation on Myc-WT Raptor in a PKA-dependent manner (blocked by H89), but liraglutide fails to increase PKA motif phosphorylation on Myc-Ser[791]Ala Raptor (*Figure 2A and B*). This was also observed in β-TC3 cells, an immortalized β-cell line endogenously expressing the GLP-1R (*Figure 2C*). A high concentration of the Akt inhibitor MK-2206 (20 μM) did not affect liraglutide-induced PKA motif phosphorylation of Raptor in CHO-GLP-1R (*Figure 2A and B*) or β-TC3 (*Figure 2C*) cells. Thus, liraglutide stimulates PKA phosphorylation of Raptor at Ser[791].

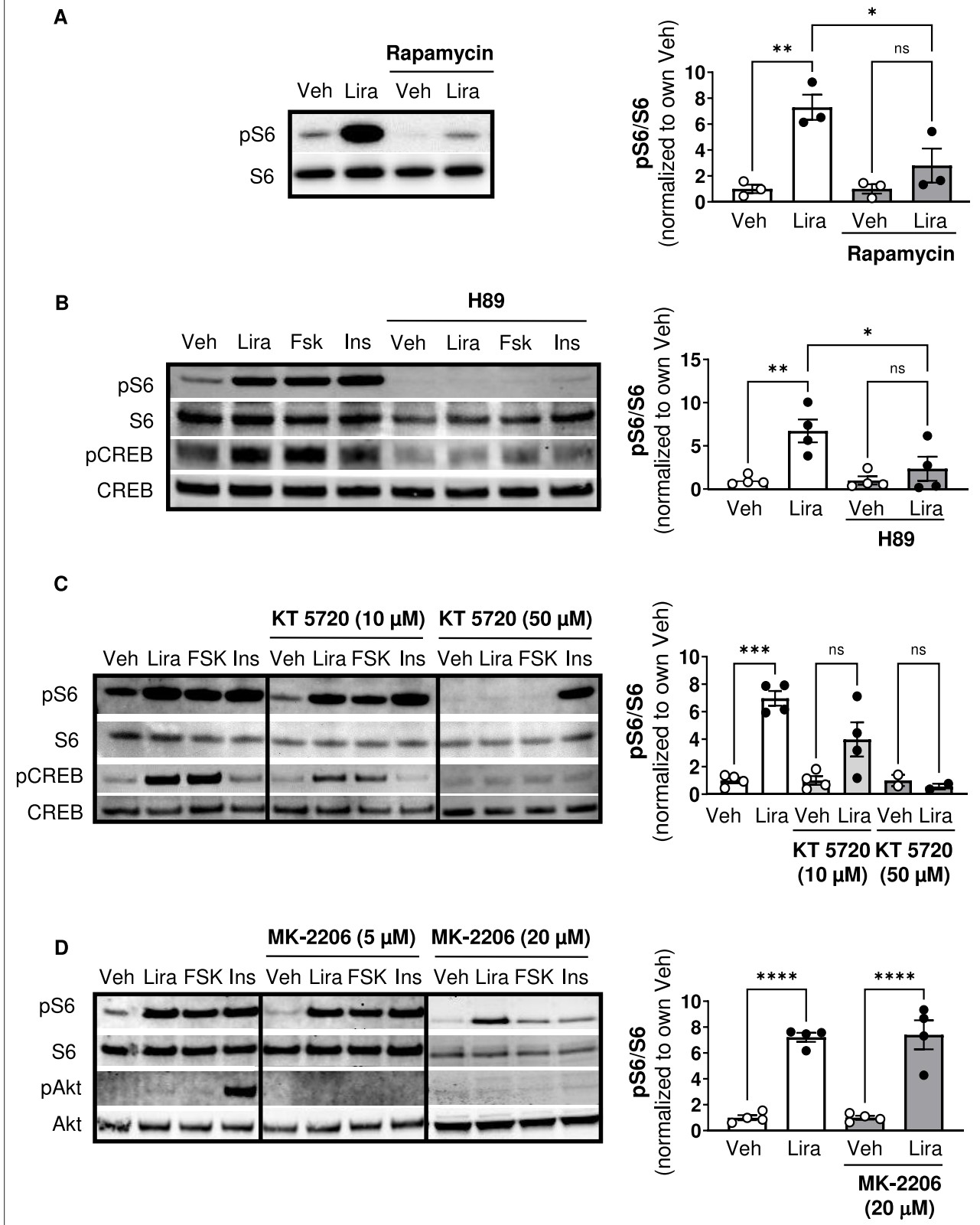

**Figure 1.** The GLP-1R agonist liraglutide increases mTOR activity in a PKA- dependent manner (Resubmission Rev 2 *Figure 1—source data 1*). (**A**) Immunoblot of pS6, S6, pCREB, and CREB and quantification of pS6 expression (pS6 /S6) in CHO-K1 cells transfected with the a vector expressing the *hGLP1R* and treated with DMSO or the mTORC1 inhibitor rapamycin (100 nM) for 30 min followed by treatment with 1 X PBS (Veh) or liraglutide (Lira, 10 nM) for 1 hr (Interaction: F (1, 8) = 6.926, p=0.0301; Inhibitor: F (1, 8) = 6.926, p=0.0301; Treatment: F (1, 8) = 22.50, p=0.0015). (**B–C**) Immunoblot

*Figure 1 continued on next page*

*Figure 1 continued*

of pS6, S6, pCREB, and CREB and quantification of pS6 expression (pS6/S6) in CHO-K1 cells stably expressing the *hGLP1R* treated with DMSO (Veh) or either of the PKA inhibitors, (**B**) H89 (10 µM) or (**C**) KT5720 (10 µM or 50 µM), for 30 min followed by treatment with vehicle (PBS), Lira (10 nM), forskolin (Fsk, 10 µM), or insulin (Ins, 10 mg/mL) for 1 hr (H89: Interaction: $F_{(1, 12)}$ = 4.782, p=0.0493; Inhibitor: $F_{(1, 12)}$ = 4.782, p=0.0493; Treatment: $F_{(1, 12)}$ = 12.50, p=0.0041). KT5720: (Interaction: $F_{(2, 18)}$ = 15.32, p=0.0001; Inhibitor: $F_{(2, 18)}$ = 15.32, p=0.0001; Treatment: $F_{(1, 18)}$ = 36.14, p<0.0001). Quantification of pCREB expression (pCREB/CREB) is shown in *Figure 1—figure supplement 1*. (**D**) Immunoblot of pS6, S6, pAkt, and Akt and quantification of pS6 expression (pS6/S6) in CHO-K1 cells stably expressing the *hGLP1R* treated with DMSO or the pan-Akt inhibitor MK-2206 (5 µM or 20 µM) for 30 min followed by treatment with vehicle (PBS), Lira (10 nM), Fsk (10 µM), or Ins (10 mg/mL) for 1 hr (Interaction: $F_{(1, 12)}$ = 0.02486, p=0.8773; Inhibitor: $F_{(1, 12)}$ = 0.02486, p=0.8773; Treatment: $F_{(1, 12)}$ = 110.2, p<0.0001). Quantification of pAkt expression (pAkt/Akt) is shown in *Figure 1—figure supplement 1*. Analysis was done using two-way ANOVA followed by Tukey's multiple comparisons. Data are normalized to the respective control (Veh or Veh and inhibitor). Absolute data are shown in *Figure 1—figure supplement 1*. All data are shown as mean ± SEM, ns not significant, * p<0.05, ** p<0.01, *** p<0.001, **** p<0.0001, n=3–4 biological replicates.

The online version of this article includes the following source data and figure supplement(s) for figure 1:

**Source data 1.** Graphs of absolute quantification of data from *Figure 1* and raw data in Excel spreadsheet.

**Figure supplement 1.** Absolute pS6, pCREB, and pAkt expression in CHO-K1 cells stably expressing the *hGLP1R* and treated with liraglutide.

**Figure supplement 1—source data 1.** Quantification of absolute values from data in *Figure 1—figure supplement 1* and raw data.

**Figure supplement 2.** Dose response and time course of absolute pS6 and pCREB expression in CHO-K1 cells stably expressing the *hGLP1R* and treated with liraglutide.

**Figure supplement 2—source data 1.** *Figure 1—figure supplement 2* and raw blots and raw data for *Figure 1—figure supplement 2*.

## PKA phosphorylation of Raptor at Ser<sup>791</sup> mediates GLP-1R agonist-induced weight loss

We tested the physiological role of PKA phosphorylation of Raptor at Ser[791] by measuring body weight and glycemic responses to liraglutide in wild-type (WT) and knock-in mice in which endogenous Raptor is replaced by a PKA-resistant Raptor expressing alanine instead of serine at position 791 in all tissues (CMV-Ser[791]Ala Raptor). WT and CMV-Ser[791]Ala Raptor knock-in mice lost weight in response to liraglutide treatment, but liraglutide-induced weight loss was attenuated in the CMV-Ser[791]Ala Raptor knock-in mice (*Figure 3A–D*). This was primarily due to a lesser reduction in fat mass in liraglutide-treated CMV-Ser[791]Ala Raptor knock-in compared to WT mice (*Figure 3E–F*). There was no significant difference between genotypes in the ability of liraglutide to reduce lean (*Figure 3G–H*) or free fluid (not shown) mass. Metabolic cage experiments demonstrate that food intake was equally suppressed in WT and CMV-Ser[791]Ala Raptor knock-in mice during the first day of liraglutide treatment. However, food intake returned to pre-liraglutide treatment levels in CMV-Ser[791]Ala Raptor knock-in mice by the second day of treatment whereas they did not return to pre-liraglutide treatment levels in WT mice until the fifth day of dosing (*Figure 3I–J*). Energy expenditure (EE) decreased in response to liraglutide in both genotypes and remained significantly lower in CMV-Ser[791]Ala Raptor knock-in mice compared to their vehicle counterparts for up to 8 days of dosing (*Figure 3K–L*). In WT mice, EE only remained lower in liraglutide-treated mice for 3 days (*Figure 3K–L*). These data suggest that phosphorylation of Raptor at Ser[791] is required for the full body weight-lowering effect of liraglutide via regulation of both food intake and EE.

There was no difference between genotypes in fasting blood glucose prior to liraglutide treatment (*Figure 3—figure supplement 1*). Two weeks of liraglutide treatment lowered fasting glucose to similar degrees in both genotypes, and there was no difference in fasting plasma insulin levels (*Figure 3—figure supplement 1*).

We fed a separate cohort of control and CMV-Ser[791]Ala Raptor knock-in mice a 60% high-fat diet (HFD) to test whether PKA phosphorylation of Raptor at Ser[791] is also required for the weight loss effect of liraglutide in the setting of obesity. Unlike lean mice, obese CMV-Ser[791]Ala mice were not significantly protected from weight loss induced by liraglutide (*Figure 4A–D*). One caveat to interpreting these results is that CMV-Ser[791]Ala mice gained significantly more weight and fat mass when fed the 60% HFD prior to liraglutide treatment compared to wild-type mice (*Figure 4C and E*). We have anecdotally observed that mice with higher baseline body weight and fat mass tend to lose more weight in response to liraglutide. We now support this by showing a positive correlation between starting body weight and liraglutide-induced weight loss in both wild-type and CMV-Ser[791]Ala mice (*Figure 4—figure supplement 1*). Although there is no difference in the relationship between

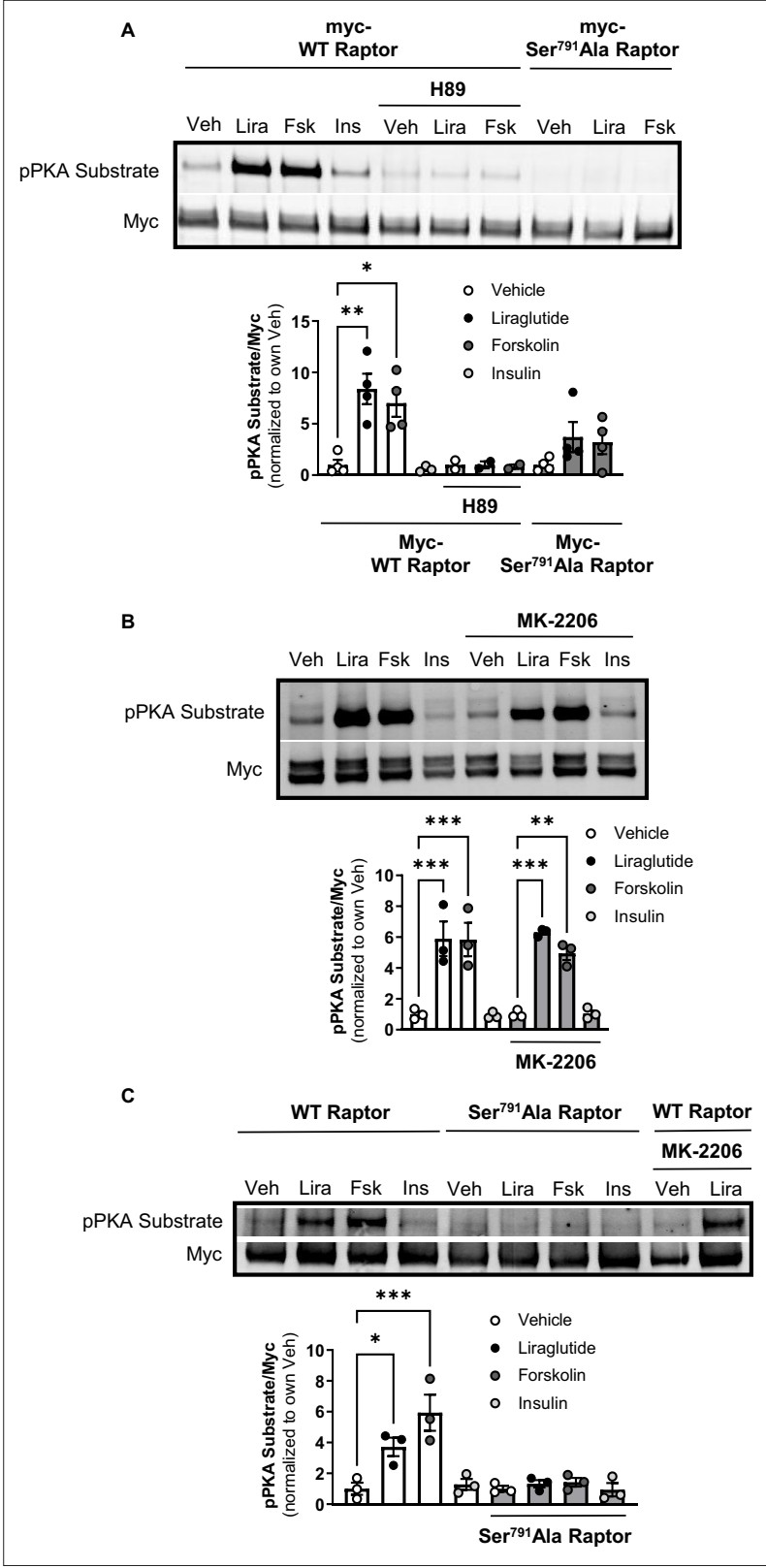

**Figure 2.** PKA phosphorylates Raptor at S791 upon GLP-1R activation (Resubmission Rev 2 *Figure 2—source data 1*). (**A**) Immunoblot and quantification of phospho-PKA substrate (pRRXS*/T*) and Myc in CHO-K1 cells stably expressing the *hGLP1R* transfected with Myc-wild-type (WT) Raptor or Myc-Ser791Ala Raptor and treated with vehicle (DMSO) or H89 (10 µM) for 30 min followed by treatment with Lira (10 nM), forskolin (Fsk, 10 µM), or

*Figure 2 continued on next page*

*Figure 2 continued*

insulin (Ins, 10 mg/mL) for 1 hr (Interaction: $F_{(4, 21)}$ = 3.695, p=0.0071; Genotype: $F_{(2, 21)}$ = 10.51, p=0.0007; Treatment: $F_{(2, 21)}$ = 6.328, p=0.0071). (**B**) Immunoblot and quantification of phospho-PKA substrate and Myc in CHO-K1 cells stably expressing the *hGLP1R* transfected with Myc-wild-type (WT) Raptor and treated with vehicle (DMSO) or MK-2206 (20 µM) for 30 min followed by treatment with Lira (10 nM), Fsk (10 µM), or Ins (10 mg/mL) for 1 hr (Interaction: $F_{(3, 16)}$ = 0.4599, p=0.7141; Inhibitor: $F_{(1, 16)}$ = 0.04511, p=0.8345; Treatment: $F_{(3, 16)}$ = 44.26, p<0.0001). (**C**) Immunoblot and quantification of phospho-PKA substrate and Myc in β-TC3 cells transfected with Myc-wild-type (WT) Raptor or Myc-Ser791Ala Raptor and treated with vehicle (DMSO) or MK-2206 (10 µM) for 30 min followed by treatment with Lira (10 nM), forskolin (Fsk, 10 µM), or insulin (Ins, 10 mg/mL) for 1 hr (Interaction: $F_{(3, 16)}$ = 7.345, p=0.0026; Genotype: $F_{(1, 16)}$ = 22.18, p=0.0002; Treatment: $F_{(3, 16)}$ = 10.93, p=0.0004). Analysis was done using two-way ANOVA followed by Tukey's multiple comparisons. Data are shown as mean ± SEM, no symbol = not significant, * p<0.05, ** p<0.01, *** p<0.001, n=3–4 biological replicates.

The online version of this article includes the following source data for figure 2:

**Source data 1.** *Figure 2*, raw blots from Figure 2, and raw data for *Figure 2*.

starting body weight and liraglutide-induced weight loss between wild-type and CMV-Ser791Ala mice (*Figure 4—figure supplement 1*), we determined the average baseline (pre-liraglutide treatment) body weight for both genotypes (43.4 g) and divided the results into mice with baseline body weights above vs. below 43.4 g (*Figure 4—figure supplement 1*). There was no difference in liraglutide-induced weight loss between wild-type and CMV-Ser791Ala mice in mice with baseline body weights above 43.4 g (*Figure 4I–J*). In mice with baseline body weights below 43.4 g, there was a tendency for CMV-Ser791Ala mice to lose less body weight in response to liraglutide (*Figure 4K–L*). These results suggest that PKA phosphorylation of Raptor may contribute to liraglutide-induced weight loss in obese mice below a certain threshold body weight.

## PKA phosphorylation of Raptor at Ser791 does not mediate melanocortin-4 receptor (MC4R) agonist-induced weight loss

To test whether Raptor phosphorylation at Ser791 is required exclusively for the weight loss effect of liraglutide or is necessary for other weight loss drugs, particularly those targeting $G_\alpha s$-coupled receptors, we treated CMV-Ser791Ala Raptor knock-in mice with the MC4R agonist setmelanotide. Like the GLP-1R, the MC4R is also a $G_\alpha s$-coupled receptor, and setmelanotide is approved for weight loss in individuals with monogenic obesity caused by mutations in pro-opiomelanocortin, proprotein convertase subtilisin/kexin type 1, or leptin receptor deficiency and in individuals with Bardet-Biedl syndrome (*Markham, 2021*). As shown in *Figure 5*, CMV-Ser791Ala Raptor knock-in mice were not resistant to weight loss in response to setmelanotide. This suggests that Raptor phosphorylation at Ser791 by PKA contributes to the weight loss effect of GLP-1R agonists but not another $G_\alpha s$-targeting pharmacological agent such as setmelanotide.

## Discussion

GLP-1R-stimulated mTORC1 signaling in the VMH reduces food intake (*Burmeister et al., 2017*). This is consistent with studies showing that activation of hypothalamic mTORC1 reduces food intake and mediates weight lowering by leptin (*Blouet et al., 2008*; *Cota et al., 2008*; *Cota et al., 2006*) . We show that PKA phosphorylates the mTORC1-regulatory protein Raptor at Ser791 in response to liraglutide and that this is responsible for ~50% of liraglutide-induced weight loss. Given the complexity of signaling mechanisms downstream of the GLP-1R (*Fletcher et al., 2016*), assigning specific pathways to a desirable phenotype could be leveraged towards more effective weight loss therapeutics.

To our knowledge, our study is the first to describe a PKA-mTORC1 interaction and its physiological relevance in the context of GLP-1R signaling. PKA is reported to either stimulate (*Liu et al., 2016*) or inhibit (*Jewell et al., 2019*) mTORC1 activity. Similar to our study, (*Liu et al., 2016*) show PKA-mediated stimulation of mTORC1 in cells incubated in serum-free media. This contrasts with inhibition of mTORC1 in cells cultured in serum-supplemented media reported by *Jewell et al., 2019*. Since mTORC1 is a cellular energy sensor, PKA exerts differential actions on mTORC1 depending on nutrient availability or cellular signaling and energy status, as originally shown by *Scott and Lawrence, 1998*

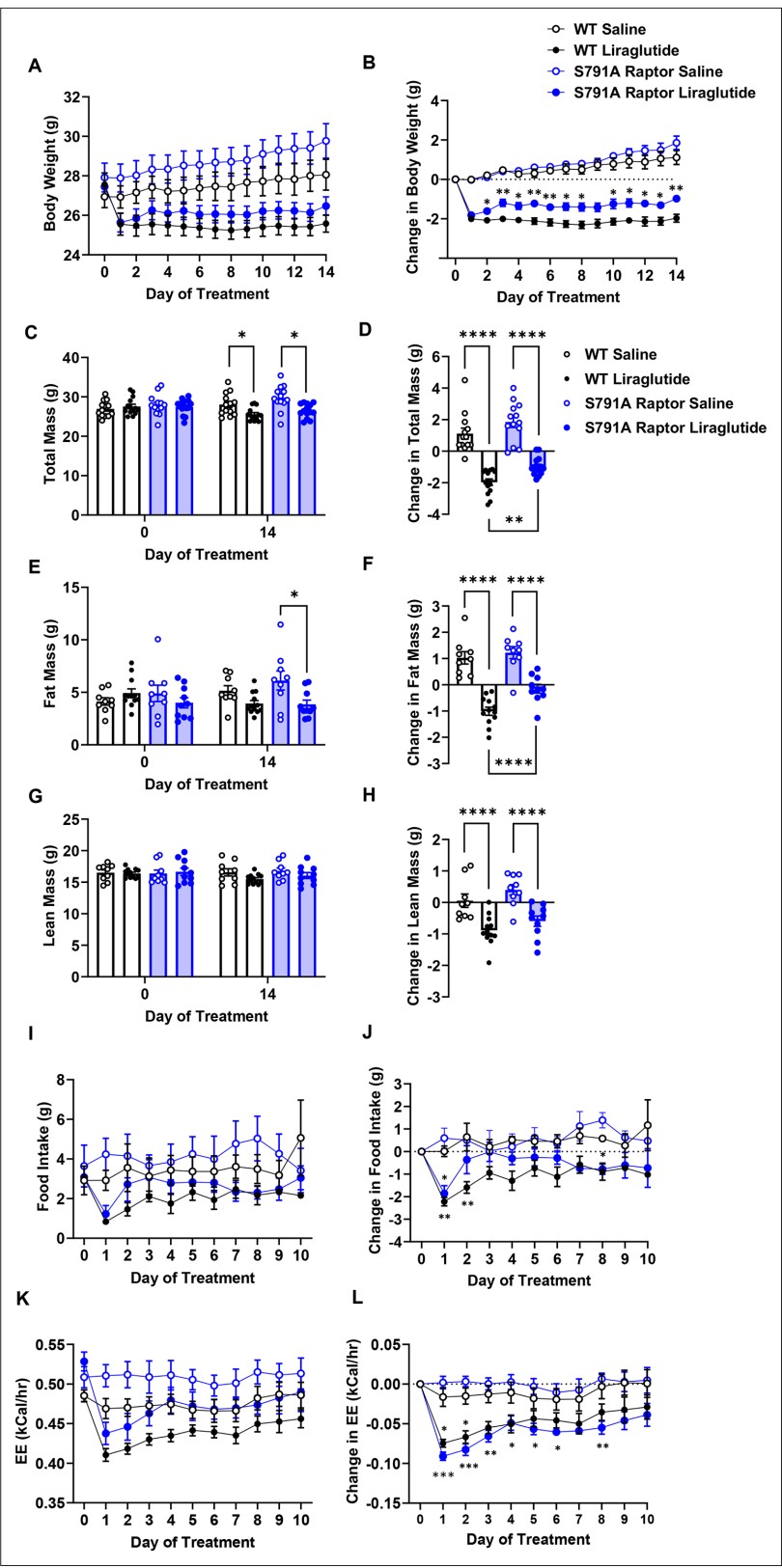

**Figure 3.** PKA phosphorylation of Raptor at S791 contributes to the weight-lowering effects of liraglutide in lean mice (Resubmission Rev 2 *Figure 3—source data 1*). Body weight, body composition, and energy balance parameters in male wild-type (WT) and CMV-Ser[791]Ala Raptor knock-in (S791A) mice given vehicle (0.9% saline) or liraglutide (200 μg/kg) subcutaneously twice a day for 14 days. Absolute body weight (**A**), total mass (**C**), fat mass

*Figure 3 continued on next page*

*Figure 3 continued*

(**E**), and lean mass (**G**). Change in body weight (**B**), total mass (**D**), fat mass (**F**), and lean mass (**H**) from baseline (Day 0). Absolute daily food intake (**I**) and EE (**J**) and change in daily food intake (**K**) and EE (**L**) relative to baseline (Day 0) during a 10-day treatment period in metabolic cages. ((**A**), Interaction: $F_{(42, 700)}=14.10$, $p<0.0001$. (**B**), Interaction: $F_{(42, 700)}=14.10$, $p<0.0001$. (**C**), Interaction: $F_{(3, 50)}=42.94$, $p<0.0001$. (**D**), $F_{(3, 50)}=42.94$, $p<0.0001$. (**E**), Interaction: $F_{(3, 36)}=30.94$, $p<0.0001$. (**F**), $F_{(3, 36)}=30.94$, $p<0.0001$. (**G**), Interaction: $F_{(3, 36)}=12.19$, $p<0.0001$. (**H**), $F_{(3, 36)}=12.19$, $p<0.0001$. (**I**), Interaction: $F_{(30, 112)}=1.370$, $p<0.0213$. (**J**), Interaction: $F_{(30, 112)}=1.706$, $p<0.0240$. (**K**), Interaction: $F_{(30, 120)}=3.827$, $p<0.0001$. (**L**), Interaction: $F_{(30, 112)}=3.827$, $p<0.0001$) Analysis was done using mixed-effects analysis followed by Holm-Sidak's multiple comparisons between liraglutide-treated WT and CMV-Ser[791]Ala Raptor mice. Data are shown as mean ± SEM, * $p<0.05$, ** $p<0.01$, **** $p<0.0001$, n=13–14 (**A**–**D**), 9–12 (**E**–**H**), or 4 (**I**–**L**) mice per group.

The online version of this article includes the following source data and figure supplement(s) for figure 3:

**Source data 1.** *Figure 3* and raw data for *Figure 3*.

**Figure supplement 1.** Blood glucose prior to drug treatment and blood glucose and insulin post-treatment in male wild-type (WT) and Ser[791]Ala Raptor knock-in mice (Resubmission Rev 2 *Figure 3—figure supplement 1—source data 1*).

**Figure supplement 1—source data 1.** *Figure 3—figure supplement 1* and raw data for *Figure 3—figure supplement 1*.

**Figure supplement 2.** Generation of CMV-Ser[791]Ala Raptor knock-in mice.

**Figure supplement 2—source data 1.** Powerpoint slide of *Figure 3—figure supplement 2*.

demonstrating that forskolin inhibits insulin-stimulated S6 phosphorylation and also subsequently shown by Liu and colleagues (*Liu et al., 2016*).

In vivo results using novel PKA-resistant Raptor knockin mice suggest that PKA phosphorylation of Raptor contributes to the weight loss effect of liraglutide in lean but not in obese mice. One limitation of these in vivo studies is that PKA-resistant Raptor is expressed in all tissues. Loss of PKA phosphorylation of Raptor in one or multiple tissues may contribute to our observation that CMV-Ser[791]Ala mice gained significantly more weight on the HFD compared to control mice. This difference in starting body weight could explain why liraglutide-induced weight loss was not attenuated in obese CMV-Ser[791]Ala mice, as in lean mice, since heavier mice tend to be more responsive to liraglutide (*Figure 4—figure supplement 1*). This could offset any reduction in liraglutide responsiveness resulting from expression of PKA-resistant Raptor. We support this by showing that CMV-Ser[791]Ala mice with lower initial body weights display a tendency towards reduced responsiveness to liraglutide. Future studies will circumvent the limitations of the CMV-Ser[791]Ala model by introducing the Ser[791]Ala mutation to specific tissues/cell types that specifically mediate liraglutide-induced weight loss. The brain is a key target for the weight-lowering effects of GLP-1R agonists (*Adams et al., 2018*; *Sisley et al., 2014*; *Varin et al., 2019*). Regulation of energy balance by mTORC1 appears to be most critical in hypothalamic regions such as the arcuate (ARC), paraventricular hypothalamus (PVH), VMH, and suprachiasmatic nucleus (*Cota et al., 2006*; *Inhoff et al., 2010*; *Proulx et al., 2008*), where the GLP-1R is highly expressed (*Jensen et al., 2018*; *Cork et al., 2015*). The ARC mediates weight loss effects of liraglutide (*Secher et al., 2014*) and is engaged by systemic liraglutide (*Salinas et al., 2018*). Hindbrain regions are also engaged by systemic liraglutide and mediate its weight-lowering effects (*Salinas et al., 2018*; *Fortin et al., 2020*). Inhibition of PKA attenuates the anorectic effect of hindbrain-targeted Ex4 in rats (*Hayes et al., 2011*). Future studies expressing Ser[791]Ala Raptor in a brain region/cell-type specific manner will enable identification of neuronal population(s) where a PKA-mTORC1 interaction mediates the anorectic effect of GLP-1R agonists.

The observation that CMV-Ser[791]Ala Raptor knock-in mice were not resistant to the weight loss effect of the MC4R agonist setmelanotide suggests that PKA-mediated phosphorylation of Raptor at Ser[791] is not a general mechanism for all $G_{\alpha}s$-coupled receptor-targeting weight loss drugs. However, as with our results using liraglutide in HFD-fed mice, a caveat to interpreting our findings with setmelanotide is the use of the whole-body CMV-Ser[791]Ala mouse model. Since the weight loss effect of setmelanotide is primarily mediated via activation of the MC4R in the PVH, future studies will test the hypothesis that the weight loss effect of setmelanotide will be blunted in PVH MC4R-specific Ser[791]Ala Raptor knock-in mice. This could establish a role for PKA-mediated regulation of mTORC1 as a key

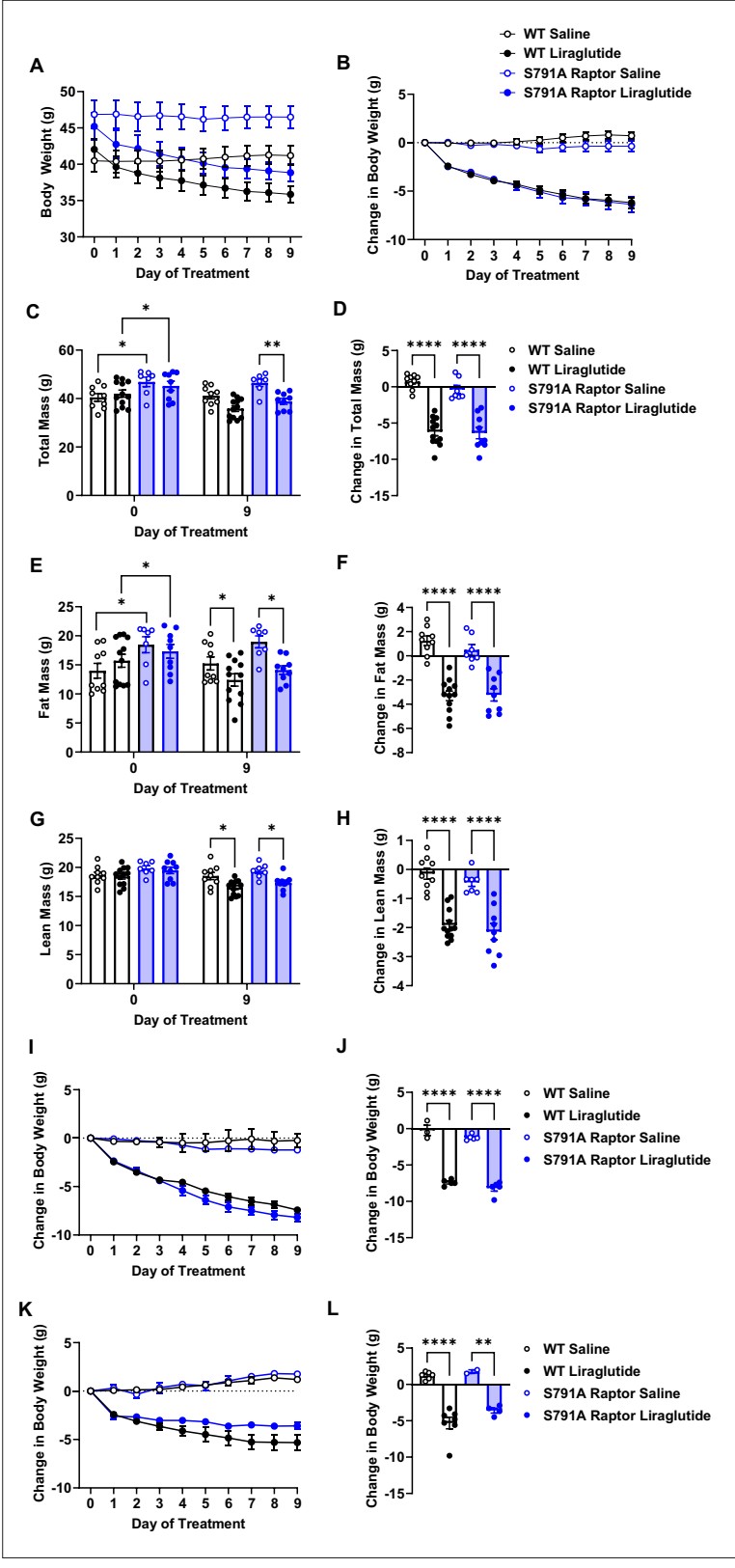

**Figure 4.** PKA phosphorylation of Raptor at S791 does not significantly contribute to the weight-lowering effects of liraglutide in obese mice (Resubmission Rev 2 *Figure 4—source data 1*). Body weight in male wild-type (WT) and CMV-Ser[791]Ala Raptor knock-in (S791A) mice fed a 60% HFD for 8–10 weeks and then given vehicle (0.9% saline) or liraglutide (200 μg/kg) subcutaneously twice a day for 9 days. Absolute body weight (**A**) and change in

*Figure 4 continued on next page*

*Figure 4 continued*

body weight (**B**) from baseline (Day 0). Absolute total mass (**C**), fat mass (**E**), and lean mass (**G**) on Days 0 and 9. Change in total mass (**D**), fat mass (**F**), and lean mass (**H**) from baseline (Day 0). Change in body weight in mice with a baseline body weight >43.4 g (**I, J**) and mice with a baseline body weight <43.4 g (**K, L**). ((**A**), Interaction: $F_{(27, 297)}=22.22$, $p<0.0001$. (**B**), Interaction: $F_{(27, 297)}=22.22$, $p<0.0001$. (**C**), Interaction: $F_{(3, 33)} = 39.82$, $p<0.0001$. (**D**), $F_{(3, 33)} = 39.82$, $p<0.0001$. (**E**), Interaction: $F_{(3, 33)} = 30.17$, $p<0.0001$. (**F**), $F_{(3, 33)} = 30.17$, $p<0.0001$. (**G**), Interaction: $F_{(3, 33)} = 25.69$, $p<0.0001$. (**H**), $F_{(3, 33)} = 25.69$, $p<0.0001$. (**I**), Interaction: $F_{(27, 126)}=40.94$, $p<0.0001$. (**J**), $F_{(3, 14)} = 131.2$, $p<0.0001$. (**K**), Interaction: $F_{(27, 135)}=17.98$, $p<0.0001$. (**L**), $F_{(3, 15)} = 29.48$, $p<0.0001$). Analysis was done using mixed-effects analysis followed by Holm-Sidak's multiple comparisons between liraglutide-treated WT and CMV-Ser[791]Ala Raptor mice. Data are shown as mean ± SEM, * $p<0.05$, ** $p<0.01$, n=7–12 mice per group (A – H), 3–5 (**I, J**), and 2–7 (**K, L**).

The online version of this article includes the following source data and figure supplement(s) for figure 4:

**Source data 1.** *Figure 4* and raw data for *Figure 4*.

**Figure supplement 1.** Relationship between baseline body weight and liraglutide-induced weight loss in male wild-type (WT) and Ser[791]Ala Raptor knock-in mice (Resubmission Rev 2 *Figure 4—figure supplement 1—source data 1*).

**Figure supplement 1—source data 1.** *Figure 4—figure supplement 1* and its raw data for *Figure 4—figure supplement 1*.

mechanism for weight loss along an ARC (target of GLP-1R agonists)-PVH (target of MC4R agonists) axis.

Our results suggest that liraglutide does not regulate mTORC1 via Akt, as was observed for βAR signaling in adipocytes (*Liu et al., 2016*). Inhibition of Akt activity in the hindbrain (*Rupprecht et al., 2013*) or hypothalamus (*Yang et al., 2017*) attenuates the anorectic effect of Ex4, so identifying targets of Akt mediating the effects of GLP-1R agonists requires further study. AMP-activated protein kinase (AMPK) is also implicated in the metabolic actions of GLP-1R agonists. Studies from our lab and others show that GLP-1R agonists reduce food intake by inhibiting AMPK (*Burmeister et al., 2017*; *Hayes et al., 2011*; *Burmeister et al., 2013*; *Sandoval et al., 2012*). Since AMPK suppresses mTORC1 by phosphorylating TSC2 (*Inoki et al., 2003*) or Raptor (*Gwinn et al., 2008*), GLP-1R activation could stimulate mTORC1 by reducing AMPK activity. AMPK can also be inhibited by mTORC1 signaling (*Dagon et al., 2012*), so GLP-1R-mediated inhibition of AMPK may be downstream of mTORC1.

In summary, we have uncovered a novel signaling mechanism downstream of the GLP-1R characterized by PKA-mediated phosphorylation of Raptor and increased mTORC1 signaling that contributes to the weight-lowering effect of liraglutide. Given the variability in response and adverse effects (e.g. nausea) of current GLP-1-based drugs, the identification of therapeutically relevant signaling profiles downstream of the GLP-1R holds promise for improving efficacy and reducing side effects of these drugs. Furthermore, our findings provide a new framework for future investigations into the significance of the PKA-mTORC1 interaction to other beneficial metabolic effects of GLP-1R agonists. GLP-1R agonists continue to be used as glucose-lowering agents for type 2 diabetes based on their insulinotropic effect via pancreatic β-cells (*Drucker, 2022*). The PKA-mTORC1 interaction could contribute to this or another β-cell phenotype associated with GLP-1R agonists (e.g. proliferation/cytoprotection). Furthermore, GLP-1R agonists also display cardioprotective effects that could be mediated directly via effects on the heart and vasculature and/or be secondary to the ability of these drugs to improve glycemia, body weight, lipid profiles, and inflammation (*Ussher and Drucker, 2023*). This provides an opportunity to investigate whether the PKA-mTORC1 interaction contributes to the cardioprotective effects of GLP-1R agonists via direct or indirect mechanisms.

# Materials and methods
## Cell culture

Chinese Hamster Ovary cells expressing human GLP-1R (CHO-GLP-1R; DiscoveRx, Fremont, CA) were maintained at 37 °C (5% $CO_2$, humidified) in Ham's F12 media containing 10 mM glucose, 1 mM pyruvate, and 1 mM glutamine (Cat#11765054, Gibco) supplemented with 10% fetal bovine serum (FBS; Cat#16140071, Gibco), 10 ml/l penicillin-streptomycin (Cat#91, Hyclone), and 0.8 mg/ml Geneticin G418 Sulfate (Cat#10131027, Gibco). β-TC3 cells (*Efrat et al., 1988*) were gifted by Dr. Richard

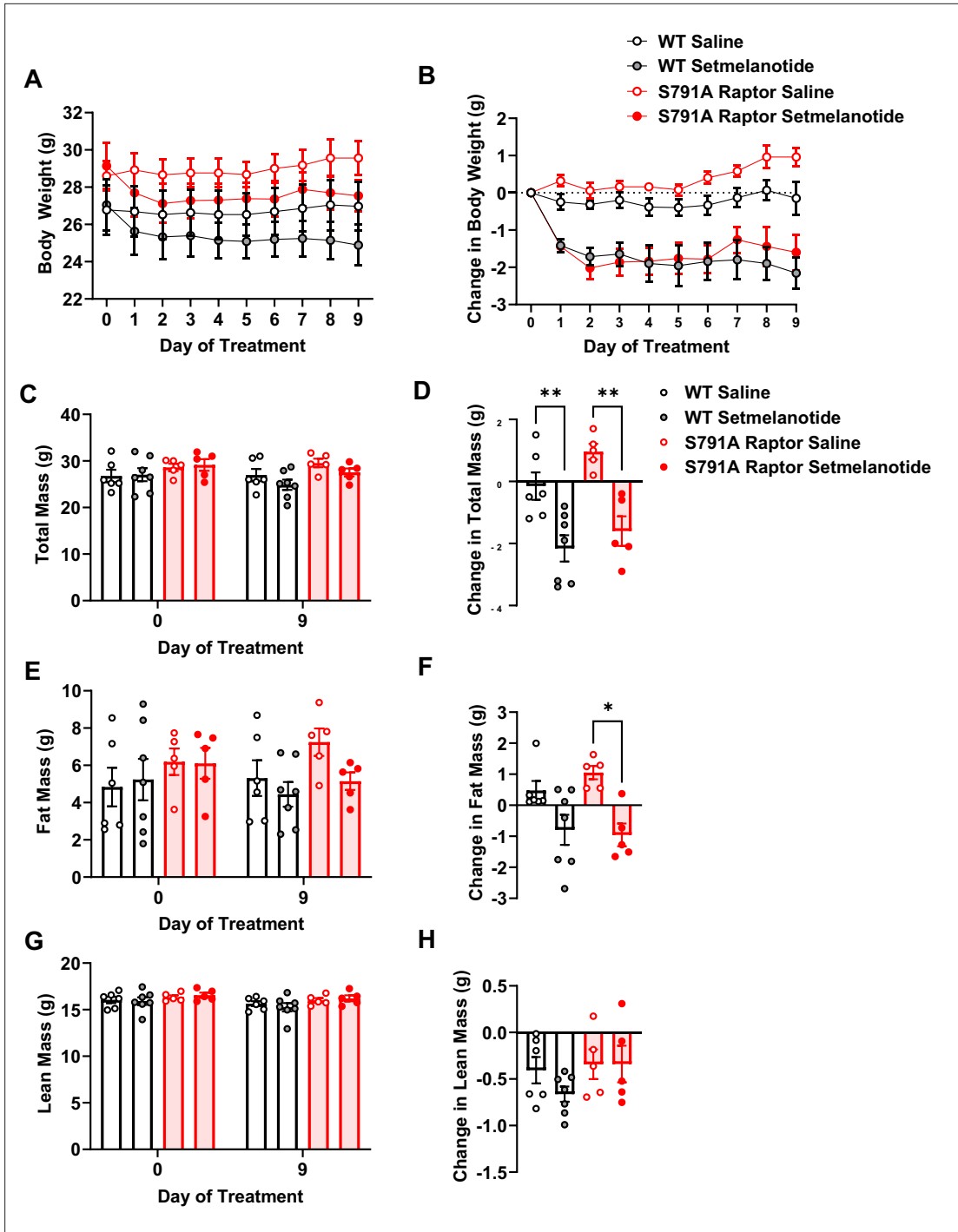

**Figure 5.** PKA phosphorylation of Raptor at S791 is not required for the weight-lowering effect of setmelanotide (Resubmission Rev 2 *Figure 5—source data 1*). Body weight and body composition in male wild-type (WT) and CMV-Ser[791]Ala Raptor knock-in (S791A) mice given vehicle (0.9% saline) or setmelanotide (4 mg/kg) subcutaneously once a day for 9 days. Absolute body weight (**A**), total mass (**C**), fat mass (**E**), and lean mass (**G**). Change in body weight (**B**), total mass (**D**), fat mass (**F**), and lean mass (**H**) from baseline (Day 0). (**A**), Interaction: $F_{(27, 171)} = 4.955$, $p < 0.0001$. (**B**), Interaction: $F_{(27, 171)} = 4.600$, $p < 0.0001$. (**C**), Interaction: $F_{(3, 19)} = 12.93$, $p < 0.0001$. (**D, F**) $_{(3, 19)} = 11.28$, $p = 0.0002$. (**E**), Interaction: $F_{(3, 19)} = 6.087$, $p = 0.0044$. (**F**), $F_{(3, 19)} = 6.087$, $p = 0.0044$. (**G**), Interaction: $F_{(3, 19)} = 1.311$, $p = 0.2999$. (**H, F**) $_{(3, 1)} = 1.311$, $p = 0.2999$. Analysis was done using mixed-effects analysis followed by Holm-Sidak's multiple comparisons between liraglutide-treated WT and CMV-Ser[791]Ala Raptor mice. Data are shown as mean ± SEM, * $p < 0.05$, ** $p < 0.01$, n=5–7 mice per group.

The online version of this article includes the following source data for figure 5:

**Source data 1.** *Figure 5* and raw data for *Figure 5*.

O'Brien and cultured in DMEM media containing 25 mM glucose, 1 mM pyruvate, and 1 mM gluta-mine (Cat#11995065, Gibco) supplemented with 15% FBS, 2% horse serum (Cat#26050088, Gibco) and 10 ml/l penicillin-streptomycin. All cells were authenticated by STR profiling (ATCC) and were verified to be free of mycoplasma. Cells were washed twice with serum-free Ham's F12 (CHO-GLP-1R) or DMEM (β-TC3) media containing 5.5 mM glucose for 3 hr and were treated with vehicle (1 X PBS) or liraglutide (Victoza, Novo Nordisk) for 1 hr with or without 30 min pre-treatment with vehicle (DMSO), PKA inhibitors H89 (Cat#2910, Tocris Bioscience) or KT5720 (Cat#1288, Tocris Bioscience), or the Akt inhibitor MK-2207 (Cat#S1078, Selleck Chemicals LLC) at concentrations stated in the figure legends. The time and dose of liraglutide treatment was determined by pilot dose response and time course experiments (*Figure 1—figure supplement 2*).

## Immunoblotting

Cells were washed once with ice-cold 1 X PBS and lysed in RIPA buffer (Cat#R0278, Sigma-Aldrich) containing 1 X protease and phosphatase inhibitors. After centrifugation of cell homogenates (17,000x*g*, 10 min, 4 °C), supernatant protein concentration was determined by BCA protein assay (Pierce BCA Protein Assay Kit, Cat#23225, Thermo Fisher Scientific). Twenty μg/lane was run (200 V, 45 min) in 4–20% Tris-glycine gels (Invitrogen) and transferred (100 V, 1 hr, 4 °C) onto PVDF membranes (Bio-Rad). Blots were blocked in buffer containing 5% BSA (Cat#A9418, Sigma-Aldrich) in 1 X Tris-Buffered Saline and 0.1% Tween–20 for 1 hr and incubated in primary antibodies (phospho-S6 Ser$^{235/236}$, Cat#4858, RRID:AB_916156; total-S6, Cat#2317, RRID:AB_2238583; p-CREB, Cat#9198, RRID:AB_2561044; total-CREB, Cat#9104, RRIS:AB_490881; phospho-Akt Thr$^{308}$, Cat#9275, RRID:AB_329828; total-Akt, Cat#9272, RRID:AB_329827; and Myc tag, Cat#2276, RRID:AB_2314825; Cell Signaling Technology) overnight at 4 °C and secondary antibodies (IRDye 800CW Goat anti-Rabbit IgG, Cat#926–32211, RRID:AB_621843 and IRDye 680RD Goat anti-Mouse IgG, Cat#926–68070, RRID:AB_10956588 LI-COR Biosciences) for 1 hr at room temperature. Blots were imaged with an Odyssey Imaging System and full lanes were analyzed using ImageJ.

## Immunoprecipitation

Myc-tagged wild-type (WT) and PKA-resistant mutant (Ser$^{791}$Ala) Raptor vectors (*Liu et al., 2016*) were transfected into CHO-GLP-1R cells using lipofectamine-3000 (Invitrogen). Forty-eight hours later, cells were washed 2 X and maintained in serum-free media for 3 hr. Cells were treated with DMSO, H89 or MK-2207 for 30 min and treated with 1 X PBS, liraglutide, forskolin, or insulin for 1 hr at concentrations stated in the figure legend. Cells were washed once with ice-cold 1 X PBS and lysed in lysis buffer (25 mM Tris HCl, 131 150 mM NaCl, 1 μM EDTA, 1 μM EGTA, 2.5 mM sodium pyrophosphate, 100 mM sodium fluoride, 1 mM sodium orthovanadate) containing 1 X protease and phosphatase inhibitors. After centrifugation of cell homogenates (17,000x*g*, 10 min, 4 °C) supernatant protein concentration was determined by BCA protein assay. A total of 200 μg protein was incubated with anti–c-Myc affinity gel (EZview Red Anti-c-Myc Affinity Gel, #E6654, Sigma-Aldrich) overnight. Beads were washed in lysis buffer and immunoblotting was done as described above using a PKA substrate monoclonal antibody (Cat#9624, Cell Signaling Technology).

## Animals

Mice were kept on a 12 hr/12 hr light/dark cycle with ad libitum access to water and chow (5L0D, LabDiet) from weaning and were studied at 12–16 weeks old. Procedures were approved by the Institutional Animal Care and Use Committee at Vanderbilt University (Protocol #M1500158). Ser$^{791}$Ala Raptor knock-in mice were generated by inGenious Targeting Laboratory Inc using the targeting vector shown in *Figure 3—figure supplement 2*. The indicated loxP sites were engineered into the genomic DNA and a two base pair point mutation was introduced into a second inverted exon 20 of the Raptor coding sequence to change the codon for Ser at position 791 to Ala. Raptor$^{fl/fl}$ mice (013188, The Jackson Laboratory) were crossed to mice expressing the Ingenious construct, and offspring were then crossed to CMV-Cre (006054, The Jackson Laboratory) mice to generate hemizygous wild-type mice or hemizygous mice with global knock-in of Raptor Ser$^{791}$Ala. This strategy was chosen because the Ingenious construct in homozygosity prior to Cre recombinase inadvertently created a hairpin loop in the transcript between the two exon 20 regions that was spliced out. Genotyping and cDNA sequencing confirmed expression of Ser$^{791}$Ala Raptor in place of endogenous

Raptor (*Figure 3*, *Figure 3—figure supplement 2*). Male mice were switched to either a low-fat diet (D12450J, Research Diets, Inc) for 4 weeks or a 60% HFD (D12492, Research Diets, Inc) for 8–10 weeks prior to initiation of dosing. Mice were maintained on the low-fat diet and were administered vehicle (0.9% saline), liraglutide (200 µg/kg body weight, BID), or setmelanotide (4 mg/kg body weight once daily) subcutaneously for 9–14 days. Body weight was measured daily, and body composition was obtained by NMR (Minispec 235 LF90II-TD NMR Analyzer, Bruker) prior to and on the last day of dosing. Four hours fasting blood glucose and insulin levels were measured using a Contour Next EZ glucose meter (Cat#BD15789, Bayer) and an Ultra Sensitive Mouse Insulin ELISA kit (Crystal Chem), respectively. A separate cohort of male mice was acclimated to a Promethion System (Sable Systems International, Inc) for one week prior to undergoing dosing with vehicle or liraglutide as described above for 10 days. Food intake and energy expenditure were measured continuously.

## Data analysis

Data were analyzed using Prism-9 (GraphPad Software, Inc). One-way or two-way ANOVA or mixed-effects analysis followed by Tukey's, Holm-Sidak's, or Brown-Forsythe multiple comparisons was used when appropriate and indicated in the figure legends. Linear regression analysis was also used as indicated in the figure legends. Sample size was determined based on the variability of the primary measurements of interest (phosphorylation of S6 for cell culture studies and body weight for in vivo studies) and the objective of detecting significance ($p < 0.05$) with 90% certainty. Treatments were randomized and blinded to the experimenter when possible.

## Acknowledgements

We thank Wei Zhang for excellent technical assistance and mouse colony management. This work was supported by NIH R01 DK132852 and DK097361 (JEA) and NIH R01 DK116625 (SC). Metabolic cage experiments were conducted by the Vanderbilt Mouse Metabolic Phenotyping Center (VMMPC) supported by NIH U2C059637 and shared instrument grant S10OD028455 (JEA). We would like to thank Merrygay James from the VMMPC for her technical assistance.

## Additional information

### Funding

| Funder | Grant reference number | Author |
| --- | --- | --- |
| National Institutes of Health | R01DK097361 | Julio E Ayala |
| National Institutes of Health | R01DK116625 | Sheila Collins |
| National Institutes of Health | S10OD028455 | Julio E Ayala |
| National Institutes of Health | R01DK132852 | Julio E Ayala |

The funders had no role in study design, data collection and interpretation, or the decision to submit the work for publication.

### Author contributions

Thao DV Le, Conceptualization, Formal analysis, Validation, Investigation, Methodology, Writing – original draft, Project administration; Dianxin Liu, Resources, Methodology; Gai-Linn K Besing, Formal analysis, Investigation, Writing - review and editing; Ritika Raghavan, Investigation; Blair J Ellis, Methodology; Ryan P Ceddia, Formal analysis; Sheila Collins, Resources, Validation, Writing - review and editing; Julio E Ayala, Conceptualization, Data curation, Supervision, Funding acquisition, Validation, Project administration, Writing - review and editing

### Author ORCIDs

Thao DV Le ⓘ https://orcid.org/0000-0002-6242-2562

Sheila Collins http://orcid.org/0000-0001-6812-8551
Julio E Ayala http://orcid.org/0000-0003-3224-2365

### Ethics

Procedures were approved by the Institutional Animal Care and Use Committee at Vanderbilt University (Protocol #M100158).

### Decision letter and Author response

Decision letter https://doi.org/10.7554/eLife.80944.sa1
Author response https://doi.org/10.7554/eLife.80944.sa2

## Additional files

### Supplementary files

• MDAR checklist

### Data availability

All data generated or analysed during this study are included in the manuscript and supporting files.

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
