## [Editor Report]

This manuscript provides support for the importance of PKA-dependent mTORC1 activation for the weight-loss effects of Liraglutide, and presumably many GPCR-dependent anorectic agents, while not affecting the food intake-reducing effects of the MC4Ractivator setmelanotide. The weight loss effect of liraglutide however was not impaired after HFD feeding, a model of obesity where GLP1 agonists are being used.

---

## [Decision Letter]

**Decision letter after peer review:**

Thank you for submitting your article "Glucagon-Like Peptide-1 Receptor Activation Stimulates PKA-Mediated Phosphorylation of Raptor and this Contributes to the Weight Loss Effect of Liraglutide" for consideration by *eLife*. Your article has been reviewed by 3 peer reviewers, and the evaluation has been overseen by a Reviewing Editor and Mone Zaidi as the Senior Editor. The following individuals involved in the review of your submission have agreed to reveal their identity: Daniela Cota (Reviewer #2); Hongxia Ren (Reviewer #3).

Essential revisions:

The reviewers agreed that the paper has potential, but the authors will need to provide substantial new data to answer the reviewers' concerns.

1) Specifically, provide a comprehensive characterization of the mouse model in addition to the studies provided.

2) Demonstrate specificity of the weight-loss effects to GPCR-PKA signaling.

3) Provide a more comprehensive analysis of the changes in energy expenditure and provide a more comprehensive metabolic phenotyping (insulin levels etc).

*Reviewer #1 (Recommendations for the authors):*

1. While I understand that the authors intend to publish a full characterization of the model elsewhere, it is impossible to evaluate the results of this manuscript without some information on the genetics of the model, and some data showing the expression of the mutant, however. The authors could add some more detail about the mouse model to this manuscript and send the manuscript describing the model in more detail for the reviewers to examine.

2. The authors should examine the normal regulation of feeding/body weight by non-GPCR-dependent agents (e.g., leptin, GDF15, or many others). They would want to examine the effects of these on PKA/pS6 in cultured cells, as well.

3. The authors should provide direct comparisons of the various conditions, rather than normalizing to individual baseline measurements.

*Reviewer #2 (Recommendations for the authors):*

1. The authors claim that the difference in efficacy in weight loss in WT vs raptor mutants treated with Liraglutide is due to a very minor difference in food intake during the second day of treatment. By looking at the data, the sustained weight difference cannot be due to a very transitory difference in food intake, which by the end of the treatment has become comparable to the food intake of vehicle-treated mice. Authors should investigate changes in energy expenditure.

2. Related to the above, previously published evidence suggests that central GLP1R agonism stimulates thermogenesis by recruiting the sympathetic nervous system (see Kooijman et al., Diabetologia 2015). It would be relevant to assess whether recruitment of the SNS is impaired in the Raptor mutant mice under liraglutide treatment. Authors could assess molecular changes for SNS markers and markers of increased thermogenesis in the brown adipose tissue and in the white adipose tissue. Authors could also assess possible differences in cold-induced SNS activity in their model.

3. It is unclear why the authors did not test the relevance of their observations by inducing diet-induced obesity in their mouse model. Does the raptor mutant have a specific phenotype under diet-induced obesity?

4. Liraglutide importantly impacts glucose metabolism. Here the authors show that 2-weeks Liraglutide treatment decreases fasting glucose levels independent of genotype. The authors however do not provide any information neither on insulin levels or on insulin sensitivity in their animal model. The mTOR pathway (including raptor) is an important funneling site of the molecular action of insulin. The fact that fasting glucose is decreased similarly in WT and mutant mice, does not preclude possible changes in fasting insulin or during dynamic tests (insulin tolerance tests).

5. Statistical analyses carried out in figure 3 do not consider as independent factors the treatment and the genotype. Authors need to carry out 2-way anovas. Only if there is an interaction, the authors can conclude that for instance the total body mass (Figure 3C) or fat mass (Figure 3E) is different between liraglutide-treated WT and mutant mice.

[Editors’ note: further revisions were suggested prior to acceptance, as described below.]

Thank you for resubmitting your work entitled "Glucagon-Like Peptide-1 Receptor Activation Stimulates PKA-Mediated Phosphorylation of Raptor and this Contributes to the Weight Loss Effect of Liraglutide" for further consideration by *eLife*. Your revised article has been evaluated by Mone Zaidi (Senior Editor) and a Reviewing Editor.

The manuscript has been improved but there are some remaining issues that need to be addressed, as outlined below:

All three reviewers and the reviewing editor feel that the revision only partly addressed the concerns raised. Specifically, the work lacks studies in diet-induced obesity which are required to explain the therapeutic mechanisms of GLP1 agonists, the failure to examine the anorectic response to other GPCR agonists to establish the specificity of the phosphorylation site for GLP1's signaling/effects versus that of other signaling pathways as the latter is what the authors imply.

*Reviewer #1 (Recommendations for the authors):*

The authors have made several useful changes to this revised manuscript, including adding additional information about the genesis of their mouse model. The validation remains a bit less complete than one would like, however- in the methods the authors state that they have sequenced the cDNA from these mice and refer to Supplemental Figure 1B, but the data are not provided here (or elsewhere).

I am also disappointed that the authors have not examined the anorectic response to other agents, as this would provide a useful and important control for the specificity of the phosphorylation site for Lira (or other GPCR agonists).

*Reviewer #2 (Recommendations for the authors):*

The authors have only partly addressed the concerns of this reviewer.

In particular, although already pointed out by this reviewer, the authors have decided to not carry out studies under diet-induced obesity. This reduces the interest for the data in their present form, which remain preliminary.

Liraglutide is used for weight loss in patients suffering from obesity and not to induce weight loss in lean individuals. The observations provided are made in non-obese mice. Whether what the authors observe still stands true in obesity and helps explain the therapeutic action of liraglutide remains to be demonstrated.

It is possible that mutant mice may have per se a phenotype in DIO, and this information would be relevant. Whether under DIO, mutant mice would respond similarly or differently from WT mice to Liraglutide would also be relevant information, even if this is a whole-body and not a cell-specific mutant, a limitation that could then be discussed by the authors.

*Reviewer #3 (Recommendations for the authors):*

The authors have made an effort to address the reviewers' comments. As a result, the quality of the manuscript was improved. However, the reviewer has the following questions to be further clarified.

1. Appropriate method of statistical analysis needs to be used for results presented in Figure 1 and the accompanying supplemental figure. The reviewer suggests to use 2-way ANOVA instead.

2. For signaling experiments using chemical probes, it is necessary to show the time course of the signaling dynamics and dose-dependent effect (as opposed to demonstrate the result using single dose and single time point).

---

## [Author Response]

Essential revisions:The reviewers agreed that the paper has potential, but the authors will need to provide substantial new data to answer the reviewers' concerns.Reviewer #1 (Recommendations for the authors):1. While I understand that the authors intend to publish a full characterization of the model elsewhere, it is impossible to evaluate the results of this manuscript without some information on the genetics of the model, and some data showing the expression of the mutant, however. The authors could add some more detail about the mouse model to this manuscript and send the manuscript describing the model in more detail for the reviewers to examine.

We gladly agree to this request and now provide detailed information about the Raptor knock-in mutant model in the Methods section (lines 206-215) and in Figure 3—figure supplement 2.

2. The authors should examine the normal regulation of feeding/body weight by non-GPCR-dependent agents (e.g., leptin, GDF15, or many others). They would want to examine the effects of these on PKA/pS6 in cultured cells, as well.

We agree with the reviewer that this is an important question that will be more appropriately addressed once we have identified the specific tissues/cell types in which Raptor signaling mediates the weight loss effect of GLP-1R agonists and generate the appropriate tissue/cell-specific Raptor mutant mouse line.

3. The authors should provide direct comparisons of the various conditions, rather than normalizing to individual baseline measurements.

We thank the reviewer for this comment. We now provide direct comparisons of the various conditions used in the cell culture experiments in Figure 1 (Figure 1—figure supplement 1) and in vivo experiments (Figure 3).

Reviewer #2 (Recommendations for the authors):1. The authors claim that the difference in efficacy in weight loss in WT vs raptor mutants treated with Liraglutide is due to a very minor difference in food intake during the second day of treatment. By looking at the data, the sustained weight difference cannot be due to a very transitory difference in food intake, which by the end of the treatment has become comparable to the food intake of vehicle-treated mice. Authors should investigate changes in energy expenditure.

we agree with the Reviewer and have now conducted an experiment in metabolic cages. As now shown in Figure 3K-N, along with a more rapid rebound in food intake in the Ser^791^Ala Raptor mutant mice during liraglutide treatment, these mutants also display reduced energy expenditure relative to vehicle-treated mice that lasts for a longer duration during the liraglutide treatment period than in wild-type mice. We hypothesize that both of these phenotypes contribute to the relative resistance to liraglutide-induced weight loss in the Raptor mutant mice.

2. Related to the above, previously published evidence suggests that central GLP1R agonism stimulates thermogenesis by recruiting the sympathetic nervous system (see Kooijman et al., Diabetologia 2015). It would be relevant to assess whether recruitment of the SNS is impaired in the Raptor mutant mice under liraglutide treatment. Authors could assess molecular changes for SNS markers and markers of increased thermogenesis in the brown adipose tissue and in the white adipose tissue. Authors could also assess possible differences in cold-induced SNS activity in their model.

Although GLP-1R agonists have been shown to increase sympathetic tone and stimulate BAT thermogenesis in rodents, it does not appear that these phenotypes contribute to parameters that regulate body weight (i.e., food intake and energy expenditure). As shown by our results in Figure 3 and studies conducted by other groups (Baggio LL et al. *Gastroenterology* 2004; Knauf C et al. *Endocrionology* 2008) GLP-1R agonist administration actually lowers energy expenditure even in wild-type mice. However, our data do suggest that the duration of liraglutide-mediated reduction of energy expenditure is greater in the Raptor mutant knock-in mice compared to control mice, even though there is no significant difference in absolute energy expenditure between genotypes. One potential interpretation of this is that mechanisms that could contribute to restoring energy expenditure (e.g. thermogenesis) are activated in control mice, but this is impaired in the Raptor mutant knock-in mice. This is a hypothesis that would be better tested in tissue-specific Raptor mutant knock-in mice since this would avoid confounding effects of impaired PKA-mediated regulation of Raptor in multiple tissues (e.g., impaired PKA-Raptor signaling in adipose tissue).

The reviewer does raise an interesting point about whether Raptor signaling downstream of the GLP-1R my play a role in the response to cold exposure. However, testing this is beyond the scope of the present studies that focus on body weight control.

3. It is unclear why the authors did not test the relevance of their observations by inducing diet-induced obesity in their mouse model. Does the raptor mutant have a specific phenotype under diet-induced obesity?

This is a valid point raised by the reviewer, and it is something that we will be testing with tissue/cell-type specific Raptor mutant mice. The concern with using high fat diet-induced obesity with the whole-body Raptor mutant line is that high fat diet may provoke secondary phenotypes in this global mouse line even prior to liraglutide treatment that would complicate interpretation of results from liraglutide treatment.

4. Liraglutide importantly impacts glucose metabolism. Here the authors show that 2-weeks Liraglutide treatment decreases fasting glucose levels independent of genotype. The authors however do not provide any information neither on insulin levels or on insulin sensitivity in their animal model. The mTOR pathway (including raptor) is an important funneling site of the molecular action of insulin. The fact that fasting glucose is decreased similarly in WT and mutant mice, does not preclude possible changes in fasting insulin or during dynamic tests (insulin tolerance tests).

We agree with the reviewer and now include data showing that in addition to no effect of liraglutide on glucose levels in the Raptor mutant mice, there is also no difference between wild-type and Raptor mutant mice on fasting insulin levels (Figure 3—figure supplement 1). We also provide data for the Reviewer in Author response image 1 showing data that glucose tolerance in response to liraglutide treatment is equally improved in wild-type and Raptor mutant mice.

**Author response image 1. sa2fig1:** 

5. Statistical analyses carried out in figure 3 do not consider as independent factors the treatment and the genotype. Authors need to carry out 2-way anovas. Only if there is an interaction, the authors can conclude that for instance the total body mass (Figure 3C) or fat mass (Figure 3E) is different between liraglutide-treated WT and mutant mice.

We thank the Reviewer for this suggestion and have re-analyzed these results using 2-way ANOVA tests. There is a significant genotype and treatment interaction with time for the various comparisons in Figure 3 that show significant differences.

[Editors’ note: what follows is the authors’ response to the second round of review.]

Reviewer #1 (Recommendations for the authors):The authors have made several useful changes to this revised manuscript, including adding additional information about the genesis of their mouse model. The validation remains a bit less complete than one would like, however- in the methods the authors state that they have sequenced the cDNA from these mice and refer to Supplemental Figure 1B, but the data are not provided here (or elsewhere).

We apologize for the confusion. The results of the sequencing validation are shown in new Figure 3—figure supplement 2. We also provide additional details for the design and creation of this mouse model in the Methods section (lines 280-283).

I am also disappointed that the authors have not examined the anorectic response to other agents, as this would provide a useful and important control for the specificity of the phosphorylation site for Lira (or other GPCR agonists).

Following this suggestion from the reviewer, we have now conducted additional experiments in which we assessed the weight loss effect of the melanocortin-4 receptor (MC4R) agonist setmelanotide in wild-type and CMV-Ser^791^Ala Raptor knockin mice. Like the Glp1r, the MC4R is a G_α_s-coupled receptor, so this provides a valuable comparison to the Glp1r. As we now show in the new Figure 5, unlike what we observed with liraglutide, CMV-Ser^791^Ala Raptor knockin mice are not resistant to the weight loss effect of setmelanotide. This suggests that the phosphorylation of Raptor by PKA at Ser-791 does not contribute to the weight loss effect of all G_α_s -coupled receptor agonists equally. These results are also described in the Results section (lines 141-152) and in the Discussion (lines 193-202).

Reviewer #2 (Recommendations for the authors):The authors have only partly addressed the concerns of this reviewer.In particular, although already pointed out by this reviewer, the authors have decided to not carry out studies under diet-induced obesity. This reduces the interest for the data in their present form, which remain preliminary.Liraglutide is used for weight loss in patients suffering from obesity and not to induce weight loss in lean individuals. The observations provided are made in non-obese mice. Whether what the authors observe still stands true in obesity and helps explain the therapeutic action of liraglutide remains to be demonstrated.It is possible that mutant mice may have per se a phenotype in DIO, and this information would be relevant. Whether under DIO, mutant mice would respond similarly or differently from WT mice to Liraglutide would also be relevant information, even if this is a whole-body and not a cell-specific mutant, a limitation that could then be discussed by the authors.

Following this suggestion from the reviewer, we have now conducted experiments treating wild-type and CMV-Ser^791^Ala Raptor knockin mice with liraglutide after having been rendered obese following feeding with a 60% high fat diet (HFD). As shown in the new Figure 4, unlike the results obtained in lean mice, obese CMV-Ser^791^Ala Raptor knockin mice were not resistant to the weight loss effect of liraglutide. However, it must be noted that the CMVSer^791^Ala Raptor knockin mice had significantly higher body weight and adiposity following the HFD exposure prior to any treatment (Figure 4C and 4E). It has been our experience that mice with higher starting body weight and adiposity tend to lose more weight in response to liraglutide. We support this with additional data in new Figure 4—figure supplement 1 showing a positive correlation between initial body weight and liraglutide-induced weight loss. This could offset any attenuation of the liraglutide effect in the CMV-Ser^791^Ala Raptor knockin mice. Therefore, we determined the average baseline body weight of all mice prior to liraglutide treatments (43.4g) and divided results into those from mice with baseline body weights above 43.4g and those with baseline body weights below 43.4g. We show that there is no difference in the weight loss effect of liraglutide between wild-type and CMV-Ser^791^Ala Raptor knockin mice that have and average baseline body weight above 43.4 g (Figure 4I, 4J); however, there is a tendency for the CMV-Ser^791^Ala Raptor knockin mice to be resistant to the weight loss effect of liraglutide when looking at mice whose baseline body weights are below 43.4g. These findings suggest that PKA phosphorylation of Raptor at Ser-791 may contribute to the weight loss effect of liraglutide in obese mice below a specific weight threshold. Results are described in the Results section (lines 121-140) and in the Discussion (lines 171-183).

Reviewer #3 (Recommendations for the authors):The authors have made an effort to address the reviewers' comments. As a result, the quality of the manuscript was improved. However, the reviewer has the following questions to be further clarified.1. Appropriate method of statistical analysis needs to be used for results presented in Figure 1 and the accompanying supplemental figure. The reviewer suggests to use 2-way ANOVA instead.

Following this recommendation from the reviewer, we have now re-analyzed these data using 2-way ANOVA.

2. For signaling experiments using chemical probes, it is necessary to show the time course of the signaling dynamics and dose-dependent effect (as opposed to demonstrate the result using single dose and single time point).

Following the recommendation from the reviewer, we show both dose response and time course experiments that were conducted in order to select the proper dose and time exposure of liraglutide for the signaling experiments (Figure 1—figure supplement 2, Methods lines 243-245). Different doses of the inhibitors are already shown in Figure 1.